# Effect of Elevated CO_2_ and Drought on Biomass, Gas Exchange and Wood Structure of *Eucalyptus grandis*

**DOI:** 10.3390/plants12010148

**Published:** 2022-12-28

**Authors:** Layssa da Silva Costa, Jasmin Vuralhan-Eckert, Jörg Fromm

**Affiliations:** Institute for Wood Biology, University of Hamburg, Leuschnerstrasse 91d, 21031 Hamburg, Germany

**Keywords:** climate change, drought stress, elevated carbon dioxide, gas exchange, vessel formation, hydraulic architecture, *Eucalyptus*

## Abstract

Juvenile *Eucalyptus grandis* were exposed to drought and elevated CO_2_ to evaluate the independent and interactive effects on growth, gas exchange and wood structure. Trees were grown in a greenhouse at ambient and elevated CO_2_ (aCO_2_, 410 ppm; eCO_2_, 950 ppm), in combination with daily irrigation and cyclic drought during one growing season. The results demonstrated that drought stress limited intercellular CO_2_ concentration, photosynthesis, stomatal conductance, and transpiration, which correlated with a lower increment in height, stem diameter and biomass. Drought also induced formation of frequent and narrow vessels accompanied by a reduction in vessel lumen area. Conversely, elevated CO_2_ increased intercellular CO_2_ concentration as well as photosynthesis, and partially closed stomata, leading to a more efficient water use, especially under drought. There was a clear trend towards greater biomass accumulation at eCO_2_, although the results did not show statistical significance for this parameter. We observed an increase in vessel diameter and vessel lumen area at eCO_2_, and, contrarily, the vessel frequency decreased. Thus, we conclude that eCO_2_ delayed the effects of drought and potentialized growth. However, results on vessel anatomy suggest that increasing vulnerability to cavitation due to formation of larger vessels may counteract the beneficial effects of eCO_2_ under severe drought.

## 1. Introduction

Atmospheric carbon dioxide [CO_2_] was around 280 ppm during the pre-industrial period [1]. However, it has continuously increased due to anthropogenic actions, reaching 419 ppm in February 2022 [2]. CO_2_ is the leading gas of the so-called greenhouse effect. Assuming a rise in CO_2_ to 450 ppm over the next few decades, model calculations indicate an average global temperature increase of 1.5 °C. As a result, shifts in precipitation patterns and increasing extreme weather events are foreseen, including intensification of droughts over large areas of the subtropics [3]. Drought is a major environmental threat concerning plant growth, directly affecting their morphological, physiological, biochemical, and molecular traits [4].

In the context of climate change, trees are expected to be majorly affected due to their slow regeneration and limited dispersal distances [5]. A range of studies put particular emphasis on the spread of emboli in the xylem network, triggering an eventual hydraulic failure and tree dieback under drought [6]. For instance, it was recently demonstrated that hydraulic deterioration is the primary pathway for drought-induced tree mortality in tropical rainforests [7]. A field study on natural populations of *Eucalyptus piperita* revealed that canopy dieback was highly linked to the occurrence of native embolism [8]. Furthermore, a rapid loss of xylem hydraulic conductance was observed in *Picea abies* during the progression of the Central European drought from 2018 [9].

Plant hydraulic architecture is apparently coordinated with stomata movement, which is a key regulator of leaf water vapor release and CO_2_ diffusion [10]. A previous investigation revealed that structural xylem modifications positively correlated with stomatal conductance (g_s_) in six Poplar genotypes subjected to low water availability [11]. Stomata sensitivity to declining leaf water potential is highly plastic and essential to prevent drastic leaf water loss prior to exceeding sustainable xylem tension thresholds and initiating embolism [10]. On the other hand, decreasing in stomatal conductance due to water deprivation widely constrains photosynthesis and assigns a major implication for cambial activity, tissue allocation and plant development [12].

Decreasing stomatal conductance is mediated by inhibition of H^+^-ATPases and activation of anion channels to reduce guard cell turgor pressure, leading to stomata closure to some extent [13]. This process is characterized by the occurrence of guard cell membrane depolarization driven by passive efflux of Cl^−^, malate^2−^ and NO_3_^−^, which induces outward-rectifying K^+^ channels and K^+^ efflux [14]. This osmotic adjustment in response to decreasing water potential is a key strategy to avoid dehydration under drought conditions [15,16].

The dynamics of drought-induced stomata closure are commonly associated with formation of small and frequent vessels across many plant species, such as *Eucalyptus* hybrids, *Arundo donax,* and *Populus* sp. [17,18,19]. It is further assumed that a reduction in conduit lumen dimension and thicker walls minimizes the impacts of drought-induced cavitation, through increasing resistance to implosion under extreme negative pressure [19,20].

Analyzing the general pattern of plant response to drought raises fundamental questions about how rising atmospheric CO_2_ concentration will affect plant physiological processes and regulation of water supply. It was previously demonstrated that elevated CO_2_ can improve, to some degree, drought tolerance in plants by increasing intercellular CO_2_ concentration (C_i_), a factor that stimulates anion channel activity to induce stomata closure, lower transpiration per unit leaf area and enhanced water use efficiency (WUE) [4,13,20,21]. Higher intercellular CO_2_ concentration is also associated with greater photosynthetic rates due to stimulation of the enzyme ribulose-1,5-biphosphate carboxylase/oxygenase (Rubisco) activity [22].

Therefore, the primary effect of elevated CO_2_ is positively correlated with biomass accumulation across many plant species [23,24]. For instance, a free-air elevated CO_2_ experiment showed an increase in biomass for *Alnus glutionosa, Fagus sylvatica* and *Betula pendula* [25]. Another study also indicated that doubled atmospheric CO_2_ stimulated plant growth by 37% [26].

Eucalypts are one of the most widely cultivated hard wood tree species in the world. The major grown species include *E. globulus*, *E. camaldulensis, E. urophylla*, *E. grandis* and *E. saligna*, and their hybrids account for 80% of the total area of the eucalypt planted forests [27]. *Eucalyptus* is economically valued and holds a wide range of applications, such as lumber, firewood, charcoal, sawdust, woodchips, forest tailings and essential oil [28,29]. Additionally, it is estimated that eucalypt plantations provide 60% of total hardwood fibers used for pulp products [30].

Given that *Eucalyptus* is grown in over 90 countries and plantations take over 20 million hectares, exposure to climate change will be high [31]. Commercially important eucalypt species show extensive climatic adaptability; however, despite eucalypt forests being considered highly productive, water shortage is a limiting factor preventing their full growth potential [31,32]. Rising CO_2_ concentration has been proven to benefit drought-stressed plants [4,33,34]. However, it cannot always be assumed that exposure ameliorates the impacts of drought, since plant sensitivity markedly varies depending on species, developmental stage, and stress intensity [35].

To date, knowledge on how increasing levels of CO_2_ interact with drought on eucalypt species is not fully understood. Therefore, this study aims to explore the morphological and physiological responses of *Eucalyptus grandis* plants to drought and elevated CO_2_, independently and in combination. We hypothesize that (1) elevated CO_2_ improves photosynthesis and leads to greater biomass accumulation in well-watered *Eucalyptus grandis*; and (2) elevated CO_2_ delays the impacts of drought in *Eucalyptus grandis* by improving water use efficiency.

## 2. Results

### 2.1. Seasonal Variations in Gas Exchange at aCO_2_ and eCO_2_

The effects of elevated CO_2_ and cyclic drought on the physiology of *Eucalyptus grandis* are shown in Figure 1. In general, drought stress reduced photosynthesis (Pn) (*p* < 0.001) throughout the growing period (Figure 1a). Elevated CO_2_ mitigated the effect of drought in the first two months of growth (*p* < 0.001), raising Pn rates by 85% in May and 26.7% in June. However, the difference was no longer significant in July and August (Figure 1a). In well-watered plants, elevated CO_2_ strongly stimulated photosynthesis in May (*p* < 0.001), raising the rates by 55.3%, which was attenuated from June to August. Photosynthesis of well-watered plants was 16% higher in July (*p* < 0.01), but it was not significantly altered in the remaining months. Overall, elevated CO_2_ significantly increased Pn on well-watered plants (*p* < 0.01) and drought-stressed plants (*p* < 0.001).

Intercellular CO_2_ concentration (Ci) was substantially limited by drought stress under both CO_2_ treatments in May and June (*p* < 0.001). An acute drought in July led to a significant increase in Ci regardless of air CO_2_ concentration (*p* < 0.01) (Figure 1b). In the end of the vegetation period (August), drought-stressed plants presented significantly lower Ci rates in comparison with well-watered plants at ambient CO_2_, while plants exhibited similar Ci values at elevated CO_2_ independently of water treatment. Well-watered plants had significantly higher rates of Ci at elevated CO_2_ in relation to the ones at ambient CO_2_ (*p* < 0.001), with minor variation during the growth season (Figure 1b). In general, elevated CO_2_ increased Ci under both water treatments (*p* < 0.001).

Drought stress caused a substantial decrease in stomatal conductance (g_s_) independently of air CO_2_ levels, and in turn, transpiration (Tr) was also reduced (Figure 1c,d). There was a slight trend towards lower g_s_ and Tr in plants at elevated CO_2_ in May and June; however, elevated CO_2_ did not affect both physiological parameters at *p* < 0.05 in these two months. Regarding well-watered plants, elevated CO_2_ initially induced significant higher rates of both g_s_ and Tr (*p* < 0.01). This response shifted in the remaining months, when Tr and g_s_ rates were strongly declined under elevated CO_2_ (*p* < 0.001) (Figure 1c,d).

### 2.2. Analysis of Water Use Efficiency at aCO_2_ and eCO_2_

Drought stress and elevated CO_2_ improved water use efficiency in May and June (*p* < 0.001). Results showed that interaction of drought with elevated CO_2_ led to the highest CO_2_ uptake per transpiration unit (*p* < 0.001). Figure 2 summarizes the seasonal differences in WUE. In the first two months of growth*,* the effect of elevated CO_2_ was predominantly stronger for drought-stressed plants (*p* < 0.001). This effect decreased over the vegetation period*,* reaching a similar response in comparison with well-watered plants at elevated CO_2_ in August. Regarding well-watered plants*;* elevated CO_2_ enhanced WUE throughout the growth season especially from June onwards (*p* < 0.001)

### 2.3. Analysis of Morphological Parameters at aCO_2_ and eCO_2_

Plants under drought stress had significantly lower height at ambient CO_2_ (*p* < 0.001) and elevated CO_2_ (*p* < 0.01) in June (Figure 3). On the other hand, elevated CO_2_ played a significant role in height increment from July, by which drought plants reached statistically similar values to well-watered plants at ambient CO_2._ At the end of the growth period, elevated CO_2_ minimized drought effects to an extent where drought-stressed plants had their heights equivalent to well-watered plants at both ambient and elevated CO_2_. Eventually, elevated CO_2_ incremented the height of drought-stressed plants by 23% relative to the ones grown at ambient CO_2_ (*p* < 0.001). Under well-watered conditions, height did not significantly differ between both CO_2_ treatments. Figure 4 illustrates the plants at the end of the growth period.

Biomass accumulation was generally lower for plants under drought stress with a significantly higher increment triggered by CO_2_ interaction, but only for roots (*p* < 0.05) (Figure 5d). Concerning total biomass, there was no statistically significant difference between both CO_2_ treatments under cyclic drought condition (Figure 5b). Although, elevated CO_2_ approximated the means of the biomass of drought-stressed plants and well-watered plants grown at ambient CO_2_. Overall, drought-stressed plants at elevated CO_2_ allocated 109% more biomass to stem and had 113% more total biomass. The maximum biomass accumulation was for well-watered plants at elevated CO_2_ (Figure 5b–d), but there was no significant difference between the means. Elevated CO_2_ increased well-watered plants’ total biomass by 31.4% and stem biomass by 24.4%. Plants under drought reached the least stem diameter at ambient CO_2_ (*p* < 0.05) (Figure 5a). Elevated CO_2_ significantly affected stem diameter increment under both water treatments (*p* < 0.05).

### 2.4. Analysis of Vessel Characteristics at aCO_2_ and eCO_2_

Vessel distribution was dissimilarly disposed in well-watered and drought-stressed replicates (Figure 6). Vessel frequency was negatively correlated with vessel diameter and vessel lumen area for all treatments. Drought stress significantly increased the number of vessels per cross-sectional area (*p* < 0.001) (Figure 7b), while reducing diameter (*p* < 0.001) (Figure 7a) and area of cell lumen (*p* < 0.001) (Figure 7c). The interaction of elevated CO_2_ and drought resulted in significantly higher volume allocation to lumen area (*p* < 0.001) and lower density of vessels in comparison to drought-stressed plants at ambient CO_2_ conditions (*p* < 0.01) (Figure 7). Regarding well-watered plants, elevated CO_2_ increased vessel size (*p* < 0.001) but did not significantly affect vessel frequency. Overall, drought stress increased vessel frequency by 119.2% at ambient CO_2_ and 59% at elevated CO_2_. Contrarily, a decline in lumen area by 30.3% and 28.5% was triggered by drought at ambient and elevated CO_2_, respectively. Under well-watered treatment, elevated CO_2_ increased lumen area by 23.5%, and reduced frequency by 22.3%.

### 2.5. Analysis of Nutrient Content in Guard Cells and Chloroplasts at aCO_2_ and eCO_2_

The concentration of phosphorus (P), chloride (Cl^−^) and potassium (K^+^) in guard cells and chloroplasts from mesophyll cells were analyzed. The results obtained through energy dispersive X-ray analysis indicated that the P and K concentration did not statistically differ for all treatments. Although, K content tended to be lower at elevated CO_2_ in both guard cells (Figure 8a) and chloroplasts (Figure 8b). The results also revealed that exposure to eCO_2_ reduced the levels of Cl^−^ in guard cells under both water treatments (Figure 8a); however, the difference was only significant under well-watered conditions (*p* < 0.05). Elevated CO_2_ reduced Cl^−^ by 49.3% in the guard cells of drought-stressed plants and 60.4% in well-watered plants. Furthermore, the combination of eCO_2_ and well-watered treatment reduced Cl^−^ and potassium K^+^ concentrations in the chloroplasts, reaching 71.4% and 36.84% reduction, respectively (Figure 8b). Drought stress reduced the P content in the chloroplasts by 52% at ambient CO_2_ and by 35.7% at elevated CO_2_, while in guard cells, the P concentration declined by 27.2% at ambient CO_2_ and by 13% at elevated CO_2_.

## 3. Discussion

Multiple research projects have shown that plant response to climate change is species-dependent. Exploring the physiological mechanisms by which Eucalypt species cope with changing environmental conditions is crucial for predicting future climatic scenarios impact on growth, photosynthesis, water use and finally the specie survivorship. Although numerous studies dealt with the growth responses of trees under eCO_2_ [36,37], knowledge is still sparse about the combined effects of eCO_2_ and drought in woody species. In the present study, young *Eucalyptus* trees were grown under ambient (410 ppm) and elevated (950 ppm) CO_2_ concentration in combination with daily irrigation and drought, for one growing season.

Results showed that photosynthesis was highly affected by cyclic drought (Figure 1a), a response triggered by the reduced diffusion of carbon dioxide (CO_2_), resulting in a lower intercellular CO_2_ concentration (Figure 1b). The reduction in photosynthesis rates was further attributed to other metabolic factors indicated by increased Ci levels from the second half of the growth season (Figure 1b). The photosynthetic mechanism includes linked pathways, in which disturbances at any level eventually limit photosynthetic capacity [38]. In the present study, the results of X-ray microanalysis revealed that the P content tended to be lower in the guard cells and chloroplasts of plants under drought (Figure 8). Decreasing leaf P content has been previously observed in rapeseed cultivars under drought too [39]. Furthermore, studies demonstrated that drought limits P acquisition by decreasing the concentration of root nutrient-uptake proteins in *Hordeum vulgare*, *Zea mays* and *Andropogon gerardii* [40]. Therefore, declining inorganic phosphate could be one cause for reducing rates of photosynthesis, since P availability is essential for the regeneration of Ribulose-1,5-biphosphate (RuBP) [41].

Elevated CO_2_ positively affected the physiological responses of plants under drought by increasing intercellular CO_2_ concentration, and maintaining levels compared to those of well-watered plants under aCO_2_ (Figure 1b). Regarding well-watered plants, the eCO_2_ effects on intercellular CO_2_ were only transiently reflected in the stimulation of photosynthesis, restricted to May and July (Figure 1a). The lower effect of elevated CO_2_ for well-watered plants could be explained on the basis that acclimation to increased intercellular CO_2_ concentration occurred. Photosynthetic acclimation was previously observed on *Eucalyptus macrorhyncha* and *Eucalyptus rossii* seedlings grown at elevated CO_2_ and proper soil moisture, and was associated with increased nonstructural carbohydrates [42]. An accumulation of sugars in leaves has also caused down regulation of Rubisco in *Triticum aestivum*, leading to photosynthetic acclimation to elevated CO_2_ [43]. Another limiting factor for photosynthesis at elevated CO_2_ may include a higher nutrient level demanded by trees. Increasing atmospheric CO_2_ levels were recently reported to correlate with decreasing foliar concentration of minerals in European forests [44]. In addition, photosynthesis limitation could either occur because the plant growth exceeded the pot threshold.

Nevertheless, higher levels of intercellular CO_2_ induced the stomata to close partially. This physiological adjustment was indicated by lower rates of stomatal conductance and transpiration (Figure 1c,d), as well as reduced K^+^ and Cl^−^ content in the guard cells (Figure 8a), suggesting that a likely activation of S-type anion channels and membrane depolarization occurred. Additionally, there was a clear tendency of enhancing efficiency in water use, which was another key factor associated with CO_2_ enrichment, leading to improved growth in *E. grandis*. Consistent with that, previous investigations found that elevated CO_2_ reduced transpiration and increased water use efficiency in soybean [45], and in three woody species from the Brazilian Cerrado [46], which was particularly associated with greater biomass accumulation under the condition of sufficient watering. Our results showed that drought plants used water more efficiently in comparison with well-watered plants under our experimental approach, alike to responses observed in maize and sorghum [21]. This result denoted that elevated CO_2_ played an essential role in improving water use efficiency in virtue of decreased stomatal opening, which together with a maintenance of higher Ci levels, protected the photosynthetic apparatus and minimized the drought stress impacts on *E. grandis*. Still, plants under regular irrigation and increased CO_2_ interaction tended to have the highest biomass accumulation, showing that the effect of elevated CO_2_ in potentializing plant yield intrinsically relies on suitable soil water availability.

Because plant growth is a result of cell development, drought- and elevated-CO_2_-induced morphological responses were intimately linked with hydraulics adjustments in *E. grandis* (Figure 7). Reducing water potential and turgor pressure are directly affected by water limitation, compromising the elongation of growing cells. On the other hand, cell division appears to be less sensitive to water shortage [47]. Therefore, increasing vessel frequency and decreasing vessel diameter is a typical response to low soil moisture. In fact, our results showed that drought stress greatly increased vessel frequency, and, by contrast, decreased vessel diameter and lumen area (Figure 7). Similar responses were observed for anatomical traits of *E. grandis* and its hybrids, as well as poplar grown under water shortage [17,18]. Nevertheless, increasing the number of vessel elements could be interpreted as a strategy towards hydraulic safety, since it could either partially compensate for the lower hydraulic conductivity presumed by formation of narrower vessels, or a greater number of vessels may remain functional in case of embolism. Additionally, observations on narrower vessels suggests the presence of a lower pit area, which would reduce air seeding and increase resistance to embolism [48].

Elevated CO_2_ stimulated vessel enlargement rather than cambial division, especially when interacting with daily watering (Figure 7). This response positively correlates with the significantly higher volume allocation to the stem of *E. grandis* in relation to the control (Figure 5b). It was previously elucidated that different species distinctly respond to CO_2_ enrichment regarding vessel anatomy. For instance, no effects of elevated CO_2_ were observed in the vessel size and density of *Prunus avium*, *Betula maximowicziana* and *Acer mono*, whereas vessel size and frequency, as well as vessel lumen area, tended to increase in *Quercus robur* [49,50]. Another study showed that elevated CO_2_ decreased the size of xylem vessels and lignin deposition in legume plants [51]. The lumen size of vessels is determinant for the efficiency of water transport. There is a global pattern that associates wider vessels with improved length and higher rates of water use in trees [49]. However, this study showed that water use was declined by elevated CO_2_ while biomass tended to increase. This morphological response was supported by an enhanced hydraulic system developed under higher CO_2_ levels. Arsić et al. [52] found that *Quercus petraea* responded likewise in terms of wood structure, a modification that was accompanied by a reduction in wood density. Moreover, it was demonstrated that elevated CO_2_ may alter the chemical properties of plant cell walls differently, depending on species and CO_2_ interactions with other environmental factors [53]

The interaction of CO_2_ with drought enabled *E. grandis* to remain more turgid for a longer span, which partially mitigated the impacts of drought on hydraulics and improved growth rates. A metanalysis study showed that vessel diameter correlates positively with volume allocation to axial parenchyma, and a higher parenchyma fraction is associated with increased hydraulic conductivity and lower wood density [54]. Thusly, the trade-off between enhanced hydraulic conductance in the expense of hydraulic safety may exacerbate *E. grandis* vulnerability to embolism under increased CO_2_ levels. Conversely, refilling embolized vessels requires an efficient water transport and carbohydrate availability, both enhanced by elevated CO_2_ [55]. In future climate conditions where prolonged droughts and heat waves may occur simultaneously, the beneficial effect of elevated CO_2_ on plants is likely to decrease. This hypothesis was supported by a recent study, which evidenced that progressive drought and heat largely vanished elevated CO_2_ effects on carbon metabolism of Allepo pine [56]. Our results also demonstrated that increasing drought intensity alone (July) substantially constrained photosynthesis in *E. grandis* independently of CO_2_ enrichment (Figure 1). This response suggests that elevated CO_2_ will not offset the impacts of severe drought in *E. grandis*, at least on early developmental stages.

In conclusion, our first hypothesis was confirmed, showing that elevated CO_2_ improves photosynthesis in well-watered young *Eucalyptus grandis*. Although the total biomass did not statistically differ between irrigated plants at ambient and elevated CO_2_, we noticed a clear tendency towards greater biomass increment triggered by CO_2_ enrichment. Our second hypothesis was also confirmed, demonstrating that elevated CO_2_ delayed the effects of cyclic drought on young *Eucalyptus grandis.* The interaction of elevated CO_2_ and drought led to the highest biomass production per unit of water transpired. We observed that *Eucalyptus grandis* showed a response to drought by closing stomata to reduce dehydration. Consequently, a reduced CO_2_ diffusion into leaves, in combination with other non-stomatal factors, impaired photosynthesis and tree yield. The formation of smaller vessel elements suggests that *Eucalyptus grandis* exhibit a strategic response to improve xylem hydraulic safety and to enable survival under drought periods. Although elevated CO_2_ apparently delayed the impacts of drought stress in *Eucalyptus grandis*, it is important to consider that wider vessels could increase vulnerability to cavitation depending on duration and intervals between drought periods, which may lead to a higher risk of hydraulic failure. Wider vessels may increase vulnerability to cavitation regarding plants under adequate soil moisture, but refilling embolized vessels may be more efficient due to the CO_2_-induced lower transpiration rate. Declining stomatal conductance is the bases of predictions that trees may need less water under elevated CO_2_. Indeed, our results showed that higher CO_2_ improved water use efficiency. Although, we observed that the effect of eCO_2_ on *Eucalyptus grandis* was transient for photosynthesis, at least for plants grown in pots and environmental controlled conditions. Such a response, if realized for natural growing condition, comprises an important implication for *E. grandis* growing under future climate change scenarios. The results of this study provide understanding on how young *E. grandis* trees respond to elevated CO_2_ and drought in the short term. Despite that, knowledge on how interaction with other environmental variables is fundamental to better predict how *E. grandis* will cope with future climate change.

Finally, changes in wood structure related to water transport could interfere in terms of wood quality; particularly, forming wider vessels could impair pulp and paper production, which is the major purpose of the extensive eucalypt plantations. Further studies will focus on chemical and physical changes in wood grown under drought and/or elevated CO_2_.

## 4. Materials and Methods

### 4.1. Biological Material and Growth Conditions

The experiment was performed in the greenhouse located in Hamburg at 53°30′ N and 10°12′ E and an elevation of 25 m a.s.l., in the growing season from 2018, starting in early May and ending in early October. Climate model CC 600 (RAM co.) monitored the environmental conditions within the greenhouse chambers. The temperature in both chambers was maintained at 24 °C (±3 °C) and the relative humidity average nearing 73.6 ± 5.4%, under the photoperiod of Hamburg.

One-year-old *Eucalyptus grandis* plants were grown from seeds in pots (11 L) with uniform soil (pH 5.8) (Type ED 73, Hermann Meyer) composed of 60–80% white peat, 20–40% natural clay, nutrients (240 mg N/L, 330 mg P_2_O_3_/L, 480 mg K_2_O/L, 130 mg S/L, 160 mg Mg/L) and 2 kg/m^3^ of long-term fertilizer (Gepac LZD, Hermann Meyer, Rellingen, FRG) (Appendix A Figure A1). Twenty plants of approximately 50 cm were randomly divided in two groups of 10 individuals, one placed in the control chamber (ambient CO_2_, 410 ppm), and other placed in the elevated CO_2_ chamber (elevated CO_2_, 950 ppm). Within each chamber, plants were subdivided into two other groups of 5 individuals, and each group received a different water treatment. Plants from the well-watered group were daily watered to field saturation and plants from the drought-stressed group were cyclically irrigated. The irrigation parameter utilized in the drought-exposed plants was immediate soil rehydration to field saturation (mean 28.26 ± 1.94% (confidential level)). Soil water content was measured daily with a Theta HH2 (Delta-T devices) moisture meter. Drought-stressed plants were reirrigated every 5 days, when soil water content dropped from 28.26 ± 1.94% (confidential level) (day of irrigation) to 5.58 ± 1.04% (confidential level) (5th day post-irrigation). The treatments were applied from early May to early October.

### 4.2. Leaf Gas Exchange

Gas exchange measurements included CO_2_ uptake rate (Pn), stomatal conductance (g_s_), intercellular CO_2_ concentration (Ci) and transpiration (Tr) rate. Measurements were taken between 8:30 and 15:00, once a month using a photosynthesis system (porometer) (Li 6400 XT, Li-Cor, Biosciences, Lincoln, NE, USA) with light-emitting diode (LED) light sources (6400-02 or −02B). The measurements were performed at CO_2_ concentrations of 410 ppm and 950 ppm, relative humidity of 70% and light intensity of 1000 μmol photons m^−2^ s^−1^. The airflow through the assimilation chamber was configured to 500 μmol s^−1^. The measurements were taken from May to August, on three healthy mature leaves per plant. The leaves were acclimated for 5 min to stabilize gas exchange parameters, and data points were computed every 10 s for 1 min. IRGA’s matching was performed for every treatment in order to remove any differences between reference and samples.

Water use efficiency (WUE) was determined through mean CO_2_ uptake per mean transpiration rate.

### 4.3. Biomass Increment

Plants’ height was measured using a 2 m tape every month during the growth period. Increment in stem diameter was measured at 1 m above soil surface at the end of the growth season, using digital calipers in mm. Roots, stems and leaves were harvested at the end of the growth period, and placed at ambient temperature during 30 days for drying. Dry weight was measured in Kg at the end of the drying period. Individual values (roots and shoot) were measured separately, and total biomass was calculated.

### 4.4. EDX Microanalysis

Eucalypt leaves were shock-frozen in liquid nitrogen and freeze-dried at −110 °C for 48 h using a Heto PowerDry LL1500 freeze dryer. Leaves were subsequently broken into small pieces, cross-sectionally placed on stubs, and carbon coated under high pressure vacuum (>10^−5^ mbar) using a Bio-RAD CA508 SEM coating system. The levels of Phosphorus (P), Potassium (K^+^) and Chloride (Cl^−^) were measured in guard cells and chloroplasts from mesophyll cells using Energy Dispersive X-ray microanalysis (EDXA) on the analytical raster electron microscope Hitachi Model S520 (Hitachi Denshi, Ltd., Tokyo, Japan). In this study, the accelerate voltage was set to 10 kV and working distance was adjusted to 15 mm. The magnification range of the microscope was 20 to 200,000×.

### 4.5. Anatomical Analysis

At the end of the growth period, stem samples were collected at 10 cm above soil surface and promptly conserved in a solution of 70% ethanol. For wood structure analysis, 60 μm thick cross-sections were cut using a sliding microtome (Sartorius MI, 31 A 30), stained with Astra-blue (5%, *w*/*v*) and subsequently with safranin (1%, *w*/*v*). Slides were prepared using glycerine for further analysis of stem vessel characteristics. Vessels were observed using a light microscope (Axioskop 40, Zeiss, Göttingen, Germany) and a 2.5× objective lens equipped with a digital camera (AxioCam MRC). Whole cross-sections were screened with digital photographs (resolution: 1388 × 1040), which were merged using Adobe Photoshop CS6. For quantitative analysis, the outermost growth ring regions were selected and at least 100 vessels (diameter > 15 μm) were measured using the software Zen pro version 2012. The area of the vessels was calculated by using the following formula for elliptical calculation:A = radius a × radius b × π(1)

### 4.6. Data Analysis

The effects of the factorial water treatments (well-watered, cyclic drought) and CO_2_ concentration (410 ppm, 950 ppm) on the gas exchange and water use efficiency of *Eucalyptus grandis* were evaluated by two-way analysis of variance (ANOVA), using time interval as a random variable, at a significance level equal to or lower than 0.05 (*p* ≤ 0.05). One-way ANOVA was used to analyze the effects of the treatments on the gas exchange of Eucalyptus grandis within each month, as well as for EDXA, morphological, and wood analysis parameters. The residuals from ANOVA were tested for normality using Shapiro–Wilk’s test. Tukey’s honest significant difference (HSD) test was conducted at *p* ≤ 0.05 when there were significant differences in the F ratio from ANOVA. All statistical analyses were performed using the package “lme4” from R studio (RStudio 2022.07.2+576).

## Figures and Tables

**Figure 1 plants-12-00148-f001:**
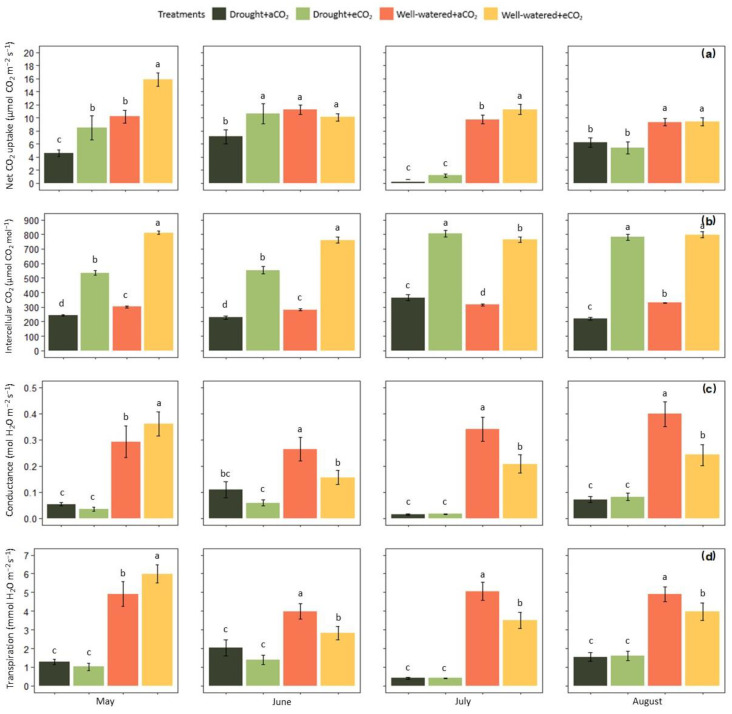
Effect of elevated CO_2_ (950 ppm) and cyclic drought stress in the mean value (±confidential level of 95%) of *Eucalyptus grandis* (**a**) CO_2_ uptake rates, (**b**) intercellular CO_2_ concentration, (**c**) stomatal conductance rates and (**d**) transpiration rates. Different small letters indicate statistical significance between the means (*p* < 0.05).

**Figure 2 plants-12-00148-f002:**
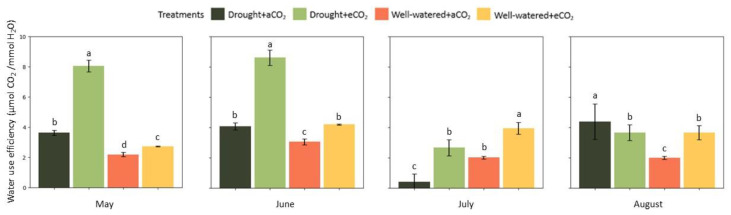
Effect of elevated CO_2_ (950 ppm) and cyclic drought stress in the mean value (±confidential level of 95%) of *Eucalyptus grandis* water use efficiency (WUE). Different small letters indicate statistical significance between the means (*p* < 0.05).

**Figure 3 plants-12-00148-f003:**
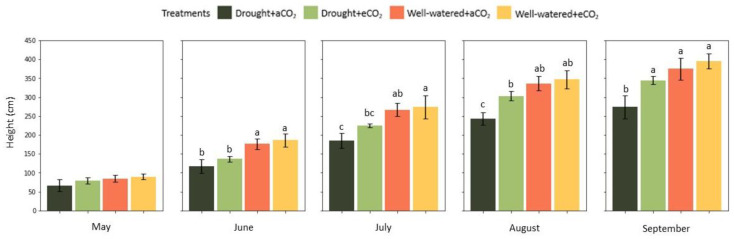
Effect of elevated CO_2_ (950 ppm) and cyclic drought stress in the mean month value (±confidential level of 95%) of *Eucalyptus grandis* height during the growth season. Different small letters indicate statistical significance between the means (*p* < 0.05).

**Figure 4 plants-12-00148-f004:**
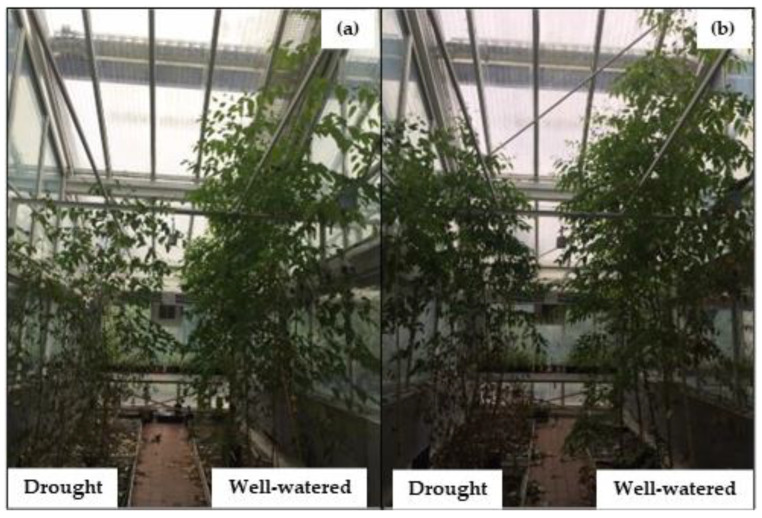
*Eucalyptus grandis* experimental plants grown at (**a**) ambient CO_2_ and (**b**) elevated CO_2_; with daily irrigation (well-watered) and cyclic drought (drought). The photograph was taken at the end of the experimental period.

**Figure 5 plants-12-00148-f005:**
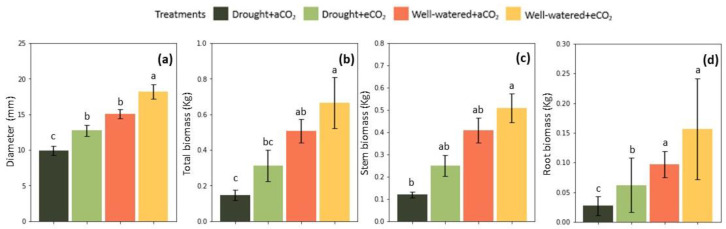
Effect of elevated CO_2_ (950 ppm) and cyclic drought stress in the mean value (±confidential level of 95%) of *Eucalyptus grandis* (**a**) stem diameter, (**b**) total biomass, (**c**)stem biomass, and (**d**) root biomass at the end of the growing season. Different small letters indicate statistical significance between the means (*p* < 0.05).

**Figure 6 plants-12-00148-f006:**
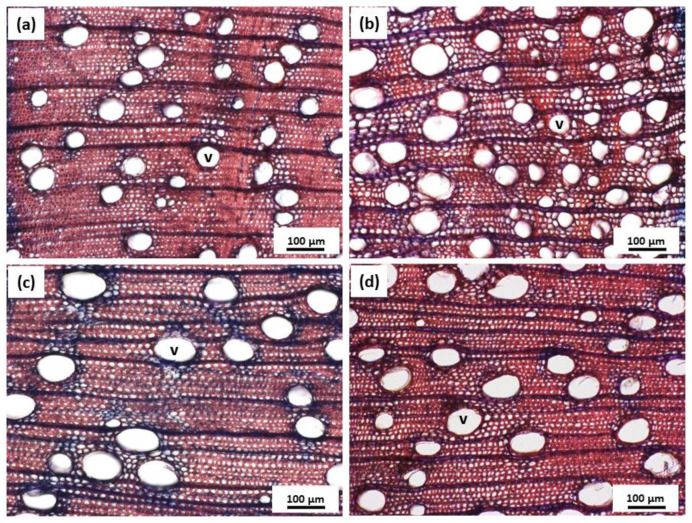
Vessel distribution of well-watered *Eucalyptus grandis* at (**a**) ambient CO_2_ and (**c**) elevated CO_2_; and drought-stressed *Eucalyptus grandis* at (**b**) ambient CO_2_ and (**d**) elevated CO_2_. V marks the vessels exemplarily. Digital photograph was taken using AxioCam MRC, Zeiss axioskop 40, 10× magnification.

**Figure 7 plants-12-00148-f007:**
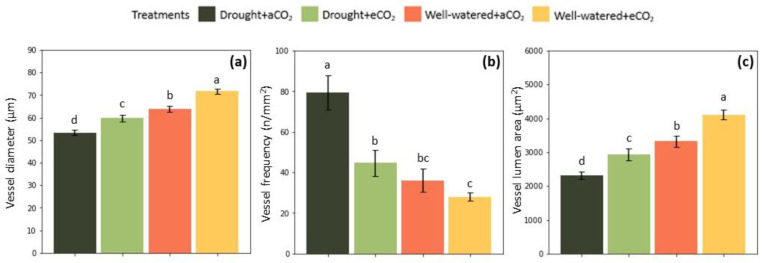
Effect of elevated CO_2_ (950 ppm) and cyclic drought stress in the mean value (±confidential level of 95%) of *Eucalyptus grandis* vessel (**a**) diameter, (**b**) frequency, and (**c**) lumen area, at the end of the growth season. Different small letters indicate statistical significance between the means (*p* < 0.05).

**Figure 8 plants-12-00148-f008:**
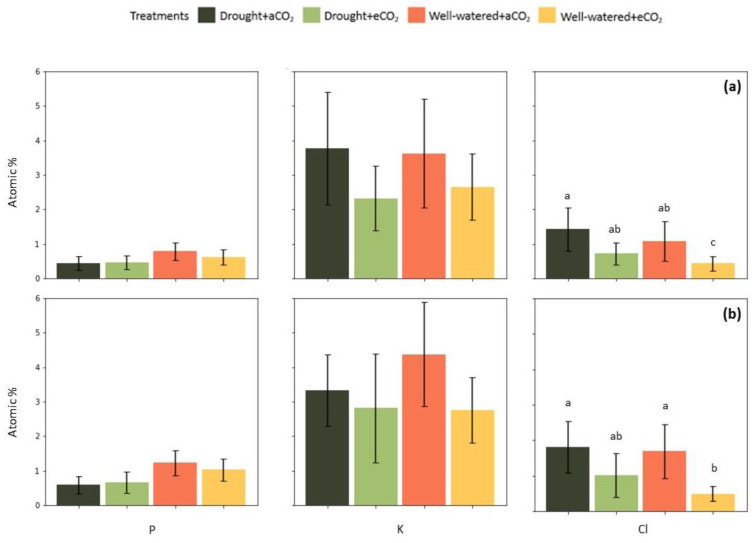
Effect of elevated CO_2_ (950 ppm) and cyclic drought stress in the mean value of EDX spectrum (±confidential level of 95%) in *Eucalyptus grandis* (**a**) guard cells and (**b**) chloroplasts of the mesophyll cells at the end of the growth season. Different small letters indicate statistical significance between the means (*p* < 0.05).

## Data Availability

The data presented in this investigation are available on request from the corresponding author.

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
