# Peer review of "Effect of Elevated CO2 and Drought on Biomass, Gas Exchange and Wood Structure of Eucalyptus grandis"

_plants, 2022, doi:10.3390/plants12010148_

Round 1
Reviewer 1 Report
The study shows the combined effect of elevated CO2 concentrations (eCO2) and drought on gas exchange, growth and wood structure in Eucalyptus grandis. The results show that drought significantly reduces Pn, gs, Ci and transpiration rate, while eCO2 have a positive effect on growth (height, stem diameter and biomass production). Drought also induced formation of frequent and narrow vessels accompanied by a reduction in vessel lumen area; whereas eCO2 increased vessel diameter and vessel lumen area and therefore reduced vessel frequency Elevated CO2 first improved growth and second delayed the effects of drought. The paper is interesting and shows good results; however the materials and methods section should be improved, the results are poorly written, must be improved. Please show the posteriori test statistics after the ANOVA of each variable, within all the graphs. I understood that the authors done Student’s T test and a value of p ≤ 0.05. I think the right analysis to be performance is ANOVA of two way (eCO2 and drought). The WUE data is important and should be shown. The paper needs to improve some aspects before being accepted for publication in the journal Plants.
Authors must give an hypothesis of the work in the manuscript
I think that when working with drought and elevated CO2 concentration an important parameter to evaluate and show is the water use efficiency. Please authors must be shown WUE
Please note that the authors should consider reducing this results session. The paragraphs in the results section are too long, I think they should be shorter and should include only the most relevant information.
Throughout the manuscript, the scientific name of the species must be in italics.
The significance level of the a posteriori test should be shown within the graphs for each parameter.
Figure 3 and Figure 8, I suggest to put it as a supplementary material.
How do they avoid the chamber effect since the experiments were done with only two cameras without replicas?
How long the plants were subjected to the treatment conditions?
Please show the results as average ± standard error (EE) instead of standard deviation (SD).
For how long were the plants subjected to each drought cycle?
What was the temperature used during the gas exchange measurements?
Author Response
Please, find the attachment.

Reviewer 2 Report
The current study explores the impact of drought and elevated CO2 on physiological and growth traits in Eucalyptus grandis grown in growth chambers. The study was done on a relevant topic, and I think it could interest readers in the field of plant ecophysiology and plant global change biology. However, I came across a few seriously methodological issues that need to be clearly addressed to be able to interpret the results of this study. First, the study was done in pots, however, I also saw that for some trees DBH was measured at 1m. Therefore, I felt that the pots may have been too small for the trees and thus there may have been some pot limitation issues on all the traits studied. However, no information was provided on how big the pots were. But looking at the picture in the method section, the pots look too small for the trees. Therefore, it would be good for the author to comment on this! Second, data were collected in different months during the growth season suggesting that some observations were not independent violating assumptions of the mode used. However, the authors did not clearly explain how they dealt with this issue. Thus, I did not continue reading the results and discussion since interpreting the results strongly depend on how good the statistical model is used to analyse the data. Last, I think authors need to provide working hypotheses on different traits studied. Once the authors have addressed these issues, I would be happy to read the revised version again.
Line 18 – 19: did not statistically differ to which treatment?
Line 34: this is an old reference! Please use the latest IPCC 2021 report.
Line 39: Please use more recent papers on this topic. Nate McDowell and colleagues have published many interesting papers that you could use to support this.
Line 43: why do you only focus on this species? I would generalize the statement.
Also add other relevant papers as well, e.g., Rowland et al. 2015. Nature.
Rowland, L. et al. Death from drought in tropical forests is triggered by hydraulics not carbon starvation. Nature 528, 119–122 (2015).
Choat, B. et al. Triggers of tree mortality under drought. Nature 558, 531–539 (2018).
Currently no working hypothesis was provided! So I would recommend adding hypotheses on different responses we may expect from your results.
Figure 9: How tall were the trees when they were placed into the treatments? Also how big (in terms of liters) were the pot used?
Line 381 – 282: the word “respectively” is out of place here. Also for clarity, these were ‘one-point’ measurements? Please specify
Line 394: hum 1 m? are you sure the pots were too small for such big trees? You need to acknowledge pot limitations in your physiological data
Line 402: at which temperature exactly?
Line 423 – 428: this section needs to be elaborated a bit more. You have taken multiple measurements throughout the growth season (May, June, July, August), meaning there was auto-correlation in your observations suggesting that you need to use statistical models that take this into account such as mixed-effect models, or nested models of some kind. You need to clearly write how this was taken into consideration. Currently its not clear to me, and also the extent to which I can trust your results/interpretation.
You may also consider using regression models in your analyses…but that properly take into account the temperal auto-correlation in your data.
Figure 1. you could either use regression analyese, or do posthoc analyses with letters indicating which treatments/month differ from each other. For now it is not clear which ones are significantly from each other. This comment goes for Figure 2, Figure 6, Figure 7
Author Response
Please, see the attachment.

Reviewer 3 Report
The manuscript “Effect of elevated CO2 and drought on biomass, gas exchange and wood structure of Eucalyptus grandis” (plants-1938968) showed a many analysis in E. gradis, which improve understand about drought on grown. Anatomical analysis, photosynthetic measurements, growth analysis, and other principal analyses were performed.
The authors have done a good work, with many relevance research’s and perceptively to Eucalyptus grown ambient and elevate CO2, which employing various references based on a scientific method and structure. The manuscript is good and minor points its necessary by adjusting in text. In addition, this manuscript demonstrated great and relevant to reader to Plants journal. I’ think, the manuscript can be accept after minor revision. Figure quality were good, and acceptable for publication.
Point:
##01: How this methodology can be extended for other plants: potential challenges, advantages? Add in your discussion and complete in conclusion topic.
-Conclusion: What are future perspectives on increased CO2 in cellulose contents by Eucalyptus plants?
-Check English sentences and syntaxes in your manuscript!
-Alphabetic order keywords;
L78 – and, not italic;
L103 – italic name;
L116 – Figure 1A;
L119 – Figure 1B; no space; check all in manuscript;
L174 – m day-1;
-Scientific names in italics; check all manuscript;
-- gs and Ci;
- Figure 1D, (mol…); legend is not correct. Check;
-Figure 2 – Not correct;
L230 – italic;
L241 – double;
Figure 7 – not correct;
L262 – not italic “and”;
Best regards,
Author Response
Please, see the attachement

Round 2
Reviewer 1 Report
In the revised version of the manuscript “Effect of Elevated CO2 and Drought on Biomass, Gas Exchange 2 and Wood Structure of Eucalyptus grandis” , I think that almost all suggestions were taken and the manuscript improved considerably. Partially included information of water use efficiency. I consider that with all the changes incorporated in the manuscript and the response to the concerns of the previous reviewers the manuscript improved considerably. I can recommend the manuscript to be published in Plants-
Minor comments
Please indicated the right units of water use efficiency, obtained by gas exchanges parameters, i.e Net CO2 uptake/transpiration rate (mmol CO2/mol H2O); if not how was determined WUE?
In references some scientific name are no write correctly, please checked it.
Author Response
Please, find the letter attached
